**Data Availability Statement:** All relevant data are within the manuscript and its Supporting information files.

# Iron status and anemia in a representative sample of US pregnant women is not associated with pre-pregnancy BMI: Results from the NHANES (1999–2010) study

Mihaela A. Ciulei[1], Kelly Gallagher[2], Djibril M. Ba[3], Celeste Beck[1], Ruth A. Pobee[4], Alison D. Gernand[1], Rachel E. Walker[1]*

1 Department of Nutritional Sciences, The Pennsylvania State University, University Park, Pennsylvania, United States of America, 2 Ross and Carol Nese College of Nursing, The Pennsylvania State University, University Park, Pennsylvania, United States of America, 3 Department of Public Health Sciences, Penn State College of Medicine, Hershey, Pennsylvania, United States of America, 4 Department of Emergency Medicine, University of Illinois at Chicago, Chicago, Illinois, United States of America

* rew5009@psu.edu

## Abstract

Iron deficiency in pregnancy is related to many poor health outcomes, including anemia and low birth weight. A small number of previous studies have identified maternal body mass index (BMI) as a potential risk factor for poor iron status. Our objective was to examine the association between pre-pregnancy BMI, iron status, and anemia in a nationally representative sample of US adult women. We used data from the National Health and Nutrition Examination Survey (NHANES; 1999–2010) for pregnant women ages 18–49 years (n = 1156). BMI (kg/m$^2$) was calculated using pre-pregnancy weight (self-reported) and height (measured at examination). Iron deficiency (ID) was defined as total body iron (calculated from serum ferritin and transferrin receptor using Cook's equation) < 0 mg/kg and anemia as hemoglobin < 11 g/dL. Associations were examined using weighted linear and Poisson regression models, adjusted for confounders (age, race/ethnicity, education, and trimester). Approximately 14% of pregnant women had ID and 8% had anemia in this sample. Ferritin and total body iron trended slightly lower (p = 0.12, p = 0.14) in women with pre-pregnancy BMI in the normal and overweight categories compared to the underweight and obese categories; hemoglobin concentrations were similar across BMI groups (p = 0.76). There were no differences in the prevalence of ID or anemia in women with pre-pregnancy overweight and obesity (ID: overweight, adjusted prevalence ratio (PR) = 1.27, 95%CI: 0.89–1.82; obesity, PR = 0.75, 95%CI: 0.39–1.45; anemia: overweight, PR = 1.08, 95%CI: 0.53–2.19; obesity, PR = 0.99, 95%CI: 0.49–2.01) compared to women with a normal BMI. Findings from these US nationally representative data indicate that total body iron, serum hemoglobin, ID, and anemia in pregnancy do not differ by pre-pregnancy BMI. Since ID and anemia during pregnancy remain significant public health concerns, NHANES should consider measuring current iron status in upcoming cycles.

**Funding:** US Department of Agriculture NIFA Postdoctoral Fellowship, grant number 2020-67034-31767 (PD: Walker).

**Competing interests:** The authors have declared that no competing interests exist.

## Introduction

During pregnancy, iron deficiency (ID) increases the risk of anemia, low birth weight, and infant mortality, while pre-pregnancy obesity increases the risk of Cesarean section, gestational diabetes, and failure to initiate breastfeeding [1–4], among other outcomes. Both ID and obesity are common in women of reproductive age in the US, especially in populations of minority race and ethnicity [5, 6].

The prevalence of ID increases from about 10% in non-pregnant women of reproductive age to about 16–25% during pregnancy due to blood volume expansion and the increased iron needs to meet fetal and placental demand [7–10], making it a significant public health concern. The severity of ID increases with higher parity [7, 8, 11] and across gestation [12, 13]. Therefore, pregnancy is a critical period for managing iron status in women of reproductive age. Based on data from the National Health and Nutrition Examination Survey (NHANES) in the US, 18% of pregnant women suffer from ID defined by total body iron [9]. However, these data are over 10 years old since NHANES has not collected biomarkers of functional and circulating iron status combined with gestational age data since 2010.

Similarly, obesity is growing in prevalence in the US population in general, including in women of reproductive age. Based on a recent analysis of national data, the prevalence of obesity in women of reproductive age was 29% in 2019 [6]. Some prior studies have suggested that women with obesity are especially vulnerable to the development of ID due to alterations in metabolism and inflammatory signaling [14, 15]. A prior review article indicated that pre-pregnancy obesity and overweight pose a risk for ID and ID with anemia (IDA) during pregnancy [16]. However, obesity is highly confounded by a number of demographic factors at a population level, making it imperative to conduct nation-wide investigations that can adjust for socio-economic variation. Given the high prevalence of pre-pregnancy obesity in the US [6] and the importance of iron to promote a healthy pregnancy, a better understanding of the relationship between these factors on a national level could help identify women at higher risk of ID.

Although there are previous studies in the US showing an association between obesity and iron status [17, 18], to our knowledge, there are no studies examining iron status and body mass index (BMI) in a US nationally representative sample of pregnant women. Previous studies were performed with a small sample size [17], in other nations [19–22], or in adolescents alone [18].

In this study, our objective was to examine iron status and anemia by pre-pregnancy BMI category in a nationally representative sample of pregnant women in the US. Based on previous studies, we hypothesized first, that women with pre-pregnancy overweight or obesity would have higher rates of ID, anemia, and IDA during pregnancy compared to women with normal weight.

## Methods

### Study population

We conducted a cross-sectional analysis of maternal pre-pregnancy BMI, iron status, and anemia using data collected in the NHANES study (https://www.cdc.gov/nchs/nhanes/index.htm). We selected the 1999–2010 cycles because they contain data on measured iron status biomarkers in addition to pregnancy status, gestational age, and self-reported pre-pregnancy weight. Additionally, these years oversampled pregnant women. In the 12 years (1999–2010; 6 cycles) that we selected, there were 62,160 participants, of which 19,569 were between the ages of 18–49 years. Next, we restricted the dataset to include only pregnant women (n = 1,414) with positive survey sampling weights (n_missing = 76), which resulted in a sample size of

1,338. We further restricted the dataset to include complete observations for measured height (n_missing = 6), self-reported pre-pregnancy weight (n_missing = 23), and iron and inflammation biomarkers (n_missing = 153). Therefore, the final analytic sample consisted of 1,156 participants. A *post hoc* power analysis was performed to determine if our sample size was sufficient to detect observed differences in prevalence using the Poisson regression option in G*Power (G*Power 3.1.9.7, Universität Düsseldorf, Germany), assuming a binomial distribution of the predictor variable. The original NHANES protocol was approved by the National Center for Health Statistics' Research Ethics Review Board [23], and we used only data publicly available for download.

## Iron and acute inflammation biomarkers

To characterize iron and inflammation status, NHANES collected maternal venous blood. Hemoglobin (Hb) was measured in whole blood, and ferritin, transferrin receptor (TfR), and the inflammatory biomarker, C-reactive protein (CRP) were measured in serum by NHANES study personnel. Serum and blood samples were collected at a single visit during the mobile examination phase of NHANES. A complete description of the protocol used in the collection of the iron and inflammatory biomarkers can be found online (https://wwwn.cdc.gov/nchs/nhanes/default.aspx).

Serum ferritin [24, 25] and TfR [26, 27] were analyzed using the immunoturbidimetric assay method via Roche kits on a Hitachi 912 clinical analyzer for 1999–2008 samples, and on an Elecsys 170 for ferritin, [28] and Hitachi Mod P for TfR [29] for the 2009–2010 samples. Due to the use of different technologies for assessment and following best-practice, we standardized ferritin concentrations using the formula, $10^{(0.989 * \log_{10}(\text{Hitachi 912})+0.049)}$ (https://wwwn.cdc.gov/Nchs/Nhanes/2009-2010/TFR_F.htm). We calculated total body iron as [log10 (TfR$_{Cook}$ *1000/*adjusted* Ferritin)-2.8229] where TfR$_{Cook}$ = *1.5* *unadjusted TfR$_{Roche}$* +0.35 to match Cook's standard values [30]. We used the total body iron cutoff < 0 mg/kg to classify ID [30]. We also defined ID as ferritin < 12 µg/L [31] and TfR > 4.4 mg/L [32]. NHANES measured Hb as part of the complete blood count analysis using a quantitative cell counter (Beckman Coulter, Brea, CA) [33, 34]. We defined anemia as Hb < 11 g/dL, per Centers for Disease Control and Prevention guidelines [31]. The American College of Obstetricians and Gynecologists suggests using a different cutoff (Hb < 10.5 g/dL) for anemia during 2nd trimester [35]. However, we chose to use a single cutoff due to the difficulty of classifying a large number of participants with 'unknown' or missing trimester data (n = 183). Instead, we have included trimester as a covariate in all adjusted models. CRP was assessed with latex-enhanced nephelometry by NHANES [36–38] (Dade Behring Inc, Deerfield, IL). We defined inflammation as CRP > 5 mg/L [39]. Ferritin is an acute phase protein and therefore can change in response to inflammation. It is recommended to adjust ferritin for inflammation with CRP and alpha(1)-acid glycoprotein using the BRINDA method in this population [40, 41]. However, NHANES did not measure alpha(1)-acid glycoprotein in the cycles we included. In addition, a linear relationship was not observed between log-transformed ferritin and log-transformed CRP, suggesting that the BRINDA method was not appropriate for this data [42]. Therefore, we did not adjust ferritin for inflammation, similar to the approach taken by previous studies [9]. IDA was defined as anemia and ID using each iron indicator separately (ferritin, TfR, and total body iron) [43].

## Body mass index

We calculated pre-pregnancy BMI (kg/m$^2$) based on pre-pregnancy weight (self-reported) and height (measured to the nearest mm in the Mobile Examination Center). We used the

following BMI categories developed by the World Health Organization: underweight ($< 18.5$ kg/m$^2$), normal (18.5–24.9 kg/m$^2$), overweight (25–29.9 kg/m$^2$), and obese ($\geq$30 kg/m$^2$) [44].

## Covariates

Maternal and household characteristics were collected via demographic and reproductive questionnaires. We selected covariates to include in our analysis that we would expect to be associated with iron status based on prior literature, making our results comparable to similar published analyses of iron status using the NHANES data [7–9, 11, 45]. We identified maternal age, race/ethnicity, maternal education, family income (NHANES calculated as poverty: income ratio) [46], and pregnancy trimester as confounders for inclusion as covariates in our models. Parity was considered but was not included due to high amounts of missing data (approximately 30%).

Maternal age and self-reported race/ethnicity variables did not have missing data. The maternal age variable was used as continuous in the analyses. The race/ethnicity variable was used as a categorical variable based on the NHANES created levels, Mexican American, Other Hispanic, non-Hispanic White, non-Hispanic Black, and Other Race–Including Multi-Racial. The 'Other' race category included participants selecting from various Asian, Pacific Islander, and Native American categories, as well as those selecting multiple races. NHANES cycles from 1999–2006 oversampled Mexican-American ethnicity, and therefore estimates for Hispanic ethnicity from those cycles represent primarily Mexican-American ethnicity (https://wwwn.cdc.gov/nchs/data/nhanes/analyticguidelines/99-10-analytic-guidelines.pdf). Due to small sample size in our current study, we combined the Mexican-American and Other Hispanic categories into one Hispanic category. However, it should be noted that the Hispanic category reliably represents only Mexican-American ethnicity from 1999–2006 (48).

There was one participant missing maternal education data, and this observation was collapsed with the high school diploma or less category, resulting in the final category of 'high school diploma or less/unknown.' Final maternal education categories were: high school diploma or less/unknown, some college, and college graduate. There were eighty-nine participants with missing family income data. These participants were coded as a separate unknown category. The final income categories were: poverty:income ratio $< 130\%$, poverty:income ratio $\geq$130, and unknown. Gestational age was self-reported as months in NHANES, and we used this information to categorize participants as first trimester (0–3 months), second trimester (4–6 months), and third trimester (7–9 months). Participants who did not know their gestational age or did not know they were pregnant were coded as unknown (n = 178). There were five participants missing gestational age data, and these observations were collapsed with participants who reported an unknown gestational age. Final trimester categories were: first trimester, second trimester, third trimester, and unknown.

## Statistical analysis

All data were analyzed using weighted statistics in SAS version 9.4 (Cary, NC) and Stata version 18 (College Station, TX), as recommended by the NHANES website (https://www.cdc.gov/nchs/nhanes/index.htm). Data merging, cleaning, and descriptive statistics were performed in SAS, while adjusted models were performed in Stata. Survey methods were used in both SAS and Stata to account for the complex sampling design of the NHANES survey, including strata, cluster, and mobile examination center weights.

Descriptive statistics, including percentages and standard errors, were conducted in the total sample and by the outcomes of interest, ID, anemia, and IDA, using weighted Chi-square tests. The iron biomarkers ferritin and TfR did not follow a normal distribution, and thus were

transformed for analyses using the natural logarithm. Total body iron and Hb followed a normal distribution and were not transformed.

For our primary analysis, associations between iron biomarker concentrations (ferritin, TfR, total body iron, and Hb; outcomes) and BMI (both continuous and categorical; exposures) were tested using weighted multiple linear regression models. Prevalence ratios and prevalence differences of ID, anemia, and IDA (outcomes) with their respective confidence intervals by BMI category (exposure) were calculated using Poisson regression models with robust error variance. All regression models were run both unadjusted and adjusted (with covariates). We considered the following variables as potential confounders: age, race/ethnicity, education level, family income, and trimester of pregnancy. We then used the hierarchical conceptual frameworks [47] to determine which covariates to include in final adjusted models. First, we considered which potential covariates were associated with iron outcomes in unadjusted models; family income was not. Second, we considered whether any covariates were located in an intermediate hierarchical level in the theoretical framework, and hypothesized family income to be between education and outcomes. Given family income had a potentially mediating relationship between education and outcomes, and was not associated with outcomes, it was not included in order to avoid modeling the same association with two different covariates. Therefore, covariates included in all final models were age, race/ethnicity, education level, and trimester of pregnancy.

When analyzing BMI as a categorical variable, the Bonferroni adjustment for multiple comparisons was used when performing pairwise comparisons. For Poisson regression models, if the number of cases was < 30 for any model, then the estimates for those analyses were deemed unreliable and noted in the results [48]. Statistical significance was defined as $p < 0.05$ for all models.

## Results

### Overall population characteristics

The mean (standard error) age for the pregnant women in this study was 28.0 (0.29) years. Sample characteristics are reported in the total sample and by ID (based on total body iron), anemia (based on Hb), and IDA (based on total body iron and Hb) in Table 1. Approximately 30% of women graduated from college and 24% had a family income below 130% of the poverty to income ratio. Overall, approximately half of the sample size had a normal pre-pregnancy BMI and were of non-Hispanic White race/ethnicity. The prevalence of ID (based on total body iron), anemia, and IDA (based on total body iron and Hb) were 14%, 8%, and 3%, respectively, and the prevalence of inflammation was 48% (Table 1). The prevalence of ID, anemia, and IDA were similar across most characteristics but were all highest during the third trimester and among non-Hispanic Black women.

### Iron status and anemia by BMI category

Ferritin and total body iron concentrations were slightly lower in women with pre-pregnancy BMI in the normal and overweight categories compared to the underweight and obese categories, but these differences were not statistically significant (p>0.10; Fig 1). There were also no statistical or meaningful differences between BMI groups in TfR or Hb. We tested if continuous pre-pregnancy BMI (kg/m$^2$) was associated with concentrations of iron biomarkers (ferritin, TfR, total body iron, and Hb), and we found no association in unadjusted or adjusted models (p>0.10; results not shown).

Similarly, we found no differences in the prevalence of ID (total body iron < 0 mg/kg), anemia (Hb < 11 g/dL), or IDA by pre-pregnancy BMI (Table 2). However, because the number

**Table 1. Characteristics of pregnant women (18–49 years of age) from NHANES 1999–2010 for the total sample and by iron deficiency, anemia, and iron deficiency anemia.**

| Variables | Total Sample N = 1156 | | Iron Deficiency[a] N = 211 | | Anemia[b] N = 100 | | IDA[c] N = 49 | |
|---|---|---|---|---|---|---|---|---|
| | N[d] | % (SE)[e] | N (cases) | % (SE) | N (cases) | % (SE) | N (cases) | % (SE) |
| **Age groups** | | | | | | | | |
| 18–24 | 428 | 31.7 (1.88) | 100 | 17.6 (3.14) | 36 | 7.65 (2.16) | 21 | 4.12 (1.59) |
| 25–29 | 352 | 28.7 (1.81) | 66 | 16.4 (3.16) | 34 | 7.36 (1.90) | 16 | 2.47 (0.88) |
| 30–34 | 258 | 23.2 (2.02) | 34 | 14.3 (3.75) | 19 | 7.40 (2.45) | 7 | 2.14 (1.10) |
| 35–49 | 118 | 16.4 (2.41) | 11 | 3.44 (1.31) | 11 | 7.88 (3.36) | 5 | 1.43 (0.71) |
| Overall p-value | | | | 0.03 | | 1.00 | | 0.31 |
| **Race/Ethnicity** | | | | | | | | |
| Hispanic | 401 | 22.3 (2.00) | 89 | 17.5 (2.56) | 38 | 8.35 (2.05) | 21 | 3.07 (0.91) |
| NH White | 512 | 53.2 (3.03) | 72 | 10.6 (1.99) | 25 | 3.28 (1.29) | 14 | 1.66 (0.93) |
| NH Black | 170 | 14.5 (1.89) | 39 | 23.3 (4.30) | 30 | 18.5 (3.81) | 12 | 7.67 (2.85) |
| NH Other | 73 | 9.97 (1.96) | 11 | 12.7 (6.94) | 7 | 12.7 (6.01) | 2 | 0.64 (0.47) |
| Overall p-value | | | | 0.09 | | < 0.001 | | 0.01 |
| **Education** | | | | | | | | |
| HS diploma or less/unknown | 597 | 41.3 (2.45) | 123 | 16.3 (2.09) | 58 | 8.70 (1.77) | 29 | 2.87 (0.74) |
| Some college | 305 | 28.9 (2.37) | 70 | 21.3 (3.95) | 29 | 8.43 (2.35) | 17 | 4.99 (2.00) |
| College graduate | 254 | 29.9 (2.54) | 18 | 4.33 (2.44) | 13 | 5.11 (2.11) | 3 | 0.41 (0.24) |
| Overall p-value | | | | 0.007 | | 0.41 | | 0.004 |
| **Family income (poverty:income ratio)** | | | | | | | | |
| < 130% | 365 | 24.2 (1.90) | 73 | 15.1 (2.35) | 40 | 10.2 (1.92) | 18 | 3.91 (1.17) |
| ≥ 130% | 702 | 68.0 (2.10) | 116 | 13.5 (1.70) | 52 | 6.68 (1.54) | 27 | 2.44 (0.83) |
| Unknown/missing | 89 | 7.80 (1.19) | 22 | 16.5 (4.63) | 8 | 7.07 (3.65) | 4 | 1.83 (1.14) |
| Overall p-value | | | | 0.72 | | 0.33 | | 0.35 |
| **Pregnancy trimester** | | | | | | | | |
| First (0–3) | 178 | 19.2 (1.88) | 9 | 2.65 (1.23) | 5 | 2.09 (1.18) | 2 | 0.98 (0.87) |
| Second (4–6) | 406 | 30.1 (2.20) | 55 | 11.2 (2.21) | 31 | 6.86 (1.84) | 16 | 2.36 (0.84) |
| Third (7–10) | 389 | 30.9 (2.21) | 111 | 23.7 (3.24) | 48 | 11.7 (2.83) | 25 | 4.07 (1.63) |
| Unknown/missing | 183 | 19.9 (1.94) | 36 | 15.0 (4.33) | 16 | 7.40 (2.21) | 6 | 2.98 (1.50) |
| Overall p-value | | | | < 0.001 | | 0.02 | | 0.38 |
| **Inflammatory Status (CRP)** | | | | | | | | |
| ≤ 5 mg/L | 551 | 52.2 (2.46) | 100 | 13.3 (1.97) | 55 | 7.53 (1.66) | 27 | 3.21 (1.06) |
| > 5 mg/L | 605 | 47.8 (2.46) | 111 | 15.1 (1.86) | 45 | 7.57 (1.67) | 22 | 2.24 (0.66) |
| Overall p-value | | | | 0.51 | | 0.99 | | 0.37 |
| **Pre-pregnancy BMI (kg/m$^2$)** | | | | | | | | |
| Underweight, < 18.5 | 66 | 5.35 (1.05) | 17 | 12.1 (4.56) | 7 | 5.14 (2.48) | 6 | 4.41 (2.33) |
| Normal, 18.5–24.9 | 609 | 52.4 (2.15) | 113 | 13.8 (2.10) | 53 | 7.05 (1.70) | 30 | 3.20 (1.04) |
| Overweight, 25.0–29.9 | 265 | 21.7 (1.74) | 49 | 18.5 (3.12) | 23 | 8.81 (2.48) | 9 | 2.63 (1.25) |
| Obese ≥ 30.0 | 216 | 20.5 (1.87) | 32 | 10.7 (3.02) | 17 | 8.11 (2.71) | 4 | 1.28 (0.85) |
| Overall p-value | | | | 0.31 | | 0.82 | | 0.48 |

[a] Iron deficiency was defined based on total body iron < 0 mg/kg. Total body iron was calculated using ferritin and transferrin receptor using Cook's equation [30].

[b] Anemia was defined based on hemoglobin < 11 g/dL.

[c] IDA was defined as both total body iron < 0 mg/kg and hemoglobin < 11 g/dL.

[d] N values were non-weighted.

[e] Percentages and standard errors for all columns were weighted and analyzed using chi-square tests.

Abbreviations: BMI = Body mass index; CRP = C-reactive protein; HS = High school; IDA = Iron deficiency anemia; NH = Non-Hispanic; SE = Standard error.

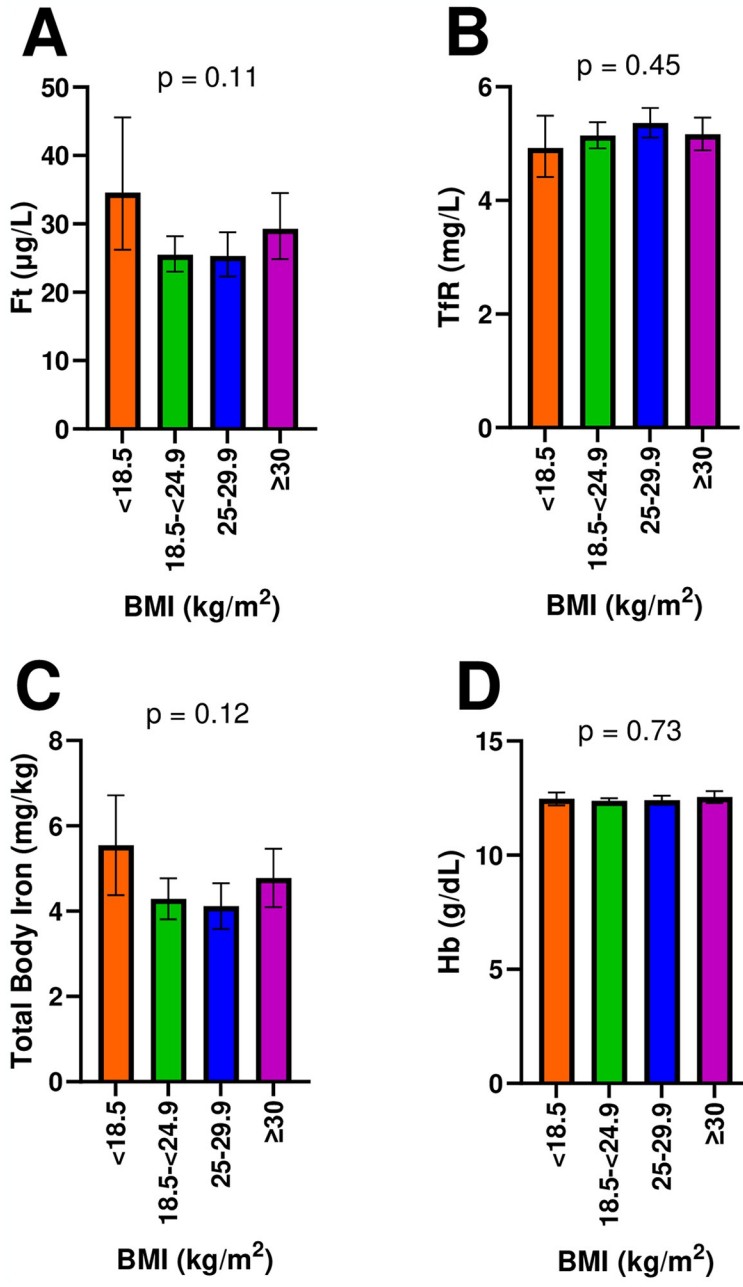

**Fig 1. Iron status biomarker concentrations by BMI category (n = 1156).** Serum concentrations of: A) Ferritin (p = 0.11), B) TfR (p = 0.45), C) Total Body Iron (p = 0.12), and D) Hb (p = 0.73) stratified by BMI categories (Underweight, < 18.5 kg/m²; Normal, 18.5–24.9 kg/m²; Overweight, 25–29.9 kg/m²; Obesity, ≥ 30 kg/m²). Data are presented as adjusted least squares means ± 95% confidence intervals (1.96*standard errors of the means). Differences between groups were assessed using weighted multiple linear regression models adjusted for age, race/ethnicity, education level, and trimester of pregnancy and adjusted for multiple comparisons using the Bonferroni method. Ferritin and transferrin receptor were log-transformed in the models to account for non-normal distributions. Means and confidence intervals were exponentiated for presentation. Total body iron was calculated using Ferritin and transferrin receptor using Cook's equation [30]. Abbreviations: BMI = Body mass index (kg/m²); Ft = Ferritin; Hb = Hemoglobin; TfR = Transferrin receptor.

**Table 2. Weighted prevalences and prevalence ratios and differences of iron deficiency and anemia status in pregnant women ages 18–49 by pre-pregnancy BMI from NHANES 1999–2010 (n = 1156).**

| Pre-pregnancy BMI categories by iron status: | Prevalence [% (Standard Error)] | Cases (N)[a] | Unadjusted Prevalence Ratio (95% CI)[b] | Adjusted Prevalence Ratio (95% CI)[c] | Adjusted Prevalence Difference [excess cases per 100 (95% CI)] |
|---|---|---|---|---|---|
| **ID (Ferritin < 12 ug/L)** | | | | | |
| Underweight BMI, < 18.5 | 15.5 (5.31) | 18 | 0.75 (0.37, 1.52)[d] | 0.78 (0.39, 1.55)[d] | -4.69 (-16.3, 6.93)[d] |
| Normal BMI, 18.5–24.9 | 20.6 (2.78) | 156 | Reference | Reference | Reference |
| Overweight BMI, 25–29.9 | 21.2 (4.18) | 64 | 1.03 (0.64, 1.64) | 0.98 (0.64, 1.50) | -0.44 (-9.31, 8.43) |
| Obese BMI, ≥ 30 | 15.5 (4.06) | 38 | 0.75 (0.41, 1.38) | 0.73 (0.41, 1.28) | -5.74 (-15.1, 3.63) |
| **ID (TfR > 4.4 mg/L)** | | | | | |
| Underweight BMI, < 18.5 | 12.5 (4.35) | 16 | 0.72 (0.34, 1.49)[d] | 0.92 (0.46, 1.84)[d] | -1.36 (-12.2, 9.46)[d] |
| Normal BMI, 18.5–24.9 | 17.4 (2.54) | 110 | Reference | Reference | Reference |
| Overweight BMI, 25–29.9 | 17.5 (2.91) | 46 | 1.01 (0.63, 1.62) | 1.07 (0.70, 1.64) | 1.21 (-6.11, 8.53) |
| Obese BMI, ≥ 30 | 14.7 (3.26) | 40 | 0.85 (0.52, 1.39) | 0.91 (0.59, 1.40) | -1.47 (-8.26, 5.31) |
| **ID (Total body iron < 0 mg/kg)[e]** | | | | | |
| Underweight BMI, < 18.5 | 12.7 (4.56) | 17 | 0.91 (0.43, 1.92)[d] | 0.88 (0.41, 1.92)[d] | -1.63 (-11.5, 8.26)[d] |
| Normal BMI, 18.5–24.9 | 13.9 (2.10) | 113 | Reference | Reference | Reference |
| Overweight BMI, 25–29.9 | 18.5 (3.12) | 49 | 1.33 (0.86, 2.05) | 1.27 (0.88, 1.82) | 3.80 (-2.06, 9.65) |
| Obese BMI, ≥ 30 | 10.7 (3.02) | 32 | 0.77 (0.40, 1.49) | 0.74 (0.39, 1.41) | -3.76 (-11.1, 3.64) |
| **Anemia (Hb < 11 g/dL)[f]** | | | | | |
| Underweight BMI, < 18.5 | 5.14 (2.48) | 7 | 0.73 (0.25, 2.13)[d] | 0.87 (0.28, 2.72)[d] | -0.99 (-8.54, 6.56)[d] |
| Normal BMI, 18.5–24.9 | 7.05 (1.70) | 53 | Reference | Reference | Reference |
| Overweight BMI, 25–29.9 | 8.81 (2.48) | 23 | 1.25 (0.61, 2.55) | 1.08 (0.53, 2.19) | 0.58 (-4.96, 6.13) |
| Obese BMI, ≥ 30 | 8.11 (2.71) | 17 | 1.15 (0.55, 2.43) | 1.00 (0.49, 2.07) | 0.03 (-5.37, 5.43) |
| **ID Anemia (Hb < 11 g/dL and Total body iron < 0 mg/kg)[f]** | | | | | |
| Underweight BMI, < 18.5 | 4.41 (2.33) | 6 | 1.38 (0.40, 4.71)[d] | 1.26 (0.36, 4.38)[d] | 0.97 (-4.70, 6.63)[d] |
| Normal BMI, 18.5–24.9 | 3.20 (1.04) | 30 | Reference | Reference | Reference |
| Overweight BMI, 25–29.9 | 2.63 (1.25) | 9 | 0.82 (0.29, 2.35) | 0.62 (0.24, 1.61) | -1.42 (-4.06, 1.22) |
| Obese BMI, ≥ 30 | 1.28 (0.85) | 4 | 0.40 (0.09, 1.70) | 0.27 (0.08, 0.90) | -2.73 (-5.07, -0.40) |

[a] N values represent non-weighted number of cases; all other analyses were weighted.

[b] Prevalence ratios and differences were calculated using Poisson regression models with robust error variance.

[c] The covariates included in all adjusted models were age, race/ethnicity, education level, and trimester of pregnancy.

[d] Estimates for the underweight BMI (< 18 kg/m$^2$) group may be unreliable because the number of cases was < 30 for all models [48].

[e] Total body iron was calculated using ferritin and transferrin receptor using Cook's equation [30].

[f] Estimates may be unreliable due to sample size of cases < 30 for some BMI categories [48].

Abbreviations: BMI = Body mass index; CI = Confidence interval; Ferritin = Ferritin; ID = Iron deficiency; IDA = Iron deficiency anemia; Hb = Hemoglobin; TfR = Transferrin receptor

of IDA cases was low within each BMI category, the estimates in adjusted models should be considered unreliable and interpreted with caution. The number of cases for all outcomes was also low in the underweight (BMI < 18.5 kg/m$^2$) category. In our *post hoc* power analysis, we examined the actual power to detect the observed prevalence ratios. Assuming an overall ID prevalence of 14% (S1 Table in S1 File), our analytical sample size of 1,156 would provide a power of 0.89–1.0 to detect prevalence ratios of 1.5–2.0, which are similar to those found for ID prevalence in other analyses (S2 Table in S1 File). We found that our sample size of 1,156 had a power of 0.73 to detect the observed ID prevalence ratio of 0.72 in the obese category

measured by serum ferritin. This sample size had a power of 0.49 to detect the prevalence ratio of 1.27 observed in the overweight category measured by total body iron. Our sample size resulted in very low power ($\leq 0.25$) to detect differences in observed prevalences of ID measured by serum TfR (overall prevalence of 16.6%), anemia (overall prevalence of 7.55%), and IDA (overall prevalence of 2.75%). Overall, any differences in prevalence by BMI category were quite mild, and our sample size was not powered to detect these small associations.

## Discussion

Based on NHANES data from 1999–2010, pre-pregnancy overweight and obesity were not associated with differences in biomarkers of iron status, including serum ferritin, serum TfR, total body iron, and hemoglobin. Additionally, pre-pregnancy overweight and obesity were not associated with increased prevalence of ID, anemia, or IDA during pregnancy in this US population. Although pre-pregnancy obesity is strongly linked to increased risk of adverse pregnancy outcomes, based on our results, it was not associated with ID or anemia. Similar to results reported by multiple previous studies [7, 9, 11, 45], we found that ID, anemia, and IDA were associated with other demographic and physiologic factors such as race/ethnicity (S2 Table in S1 File), age, education, and trimester (Table 1).

To our knowledge, the association between iron status and BMI has not been examined with nationally representative data in US pregnant women using biomarkers of either the circulating iron or storage iron components. The circulating iron component is measured with biomarkers such as TfR, serum iron, or transferrin saturation. Two prior cross-sectional cohort studies conducted in the US observed no association between circulating iron biomarkers and BMI groups [17, 18], which is similar to our findings. Of note, one of these studies [18] was conducted in adolescents (13–18 years of age) and the other [17] was limited by a small sample size (n = 15 per group). Interestingly, studies from outside of the US have found contradictory results with some finding no association between TfR and obesity [19, 22, 49] and others observing that TfR concentrations were higher (indicating poorer iron status) in women with obesity [19–21, 50].

Our results for the storage iron component, measured by ferritin, are largely consistent with other studies [20, 21, 50]. However, one study [18] found that higher concentrations of ferritin at mid-gestation and delivery were associated with classes two and three of obesity (BMI 35–39.9 and $\geq 40$ kg/m$^2$) compared with normal pre-pregnancy BMI. In the current study, we did not have a large enough sample size to stratify by obesity class. In addition, although NHANES recorded trimester at the time of assessment for each pregnant participant, there was not a large enough sample size to test for interactions between BMI category and trimester in the current study.

In addition to its function as an iron storage biomarker, ferritin is an acute phase protein, susceptible to inflammation, measured by CRP in the NHANES study. The prevalence of inflammation based on CRP >5 mg/L in our sample was 48%, which is similar to results reported by Mei et al. [9]. Previous studies have observed higher concentrations of inflammatory markers such as CRP, interleukin-6, leptin, and hepcidin in women with obesity compared with normal weight [18, 20]. In our study, pregnant women with obesity had significantly higher concentrations of the inflammatory protein, CRP, (mean (standard error) = 13.2 (1.5) mg/L) than other BMI categories (mean (standard error) = 6.6 (0.6) mg/L; p < 0.001), suggesting that elevated serum CRP is more strongly related to obesity in this sample than with acute inflammatory response. Others have found that chronic inflammation may be connected to lower ferritin concentrations through the regulatory hormone hepcidin. Hepcidin increases when iron concentrations in circulation are adequate or in the presence of

inflammation [51] and acts by blocking iron release from storage into circulation via degradation of the efflux protein, ferroportin [52]. This would suggest that obesity-related inflammation would lead to lower circulating ferritin concentrations, which was not supported by our findings. However, our results may be explained by the fact that hepcidin rises only moderately in individuals with obesity when compared to those with serious inflammatory diseases [51]. Testing this mechanism was outside the scope of our study, since NHANES has not measured hepcidin in laboratory measures. Last, the functional iron compartment, measured with Hb, was not associated with BMI categories in our sample, which is consistent with prior studies [18, 19, 21, 22].

Overall, we did not find any meaningful association with iron status biomarkers and BMI when adjusting for age, race/ethnicity, education level, and trimester of pregnancy. Although our results differed from other studies, there could be a variety of factors explaining these differences. First, distinct populations may have different associations between BMI and ID, and studies conducted in other countries may be measuring completely different populations. Second, there may be other unmeasured confounding factors such as access to medical care and food security status. For instance, access to medical care during pregnancy may delay diagnosis of ID and anemia and in turn, prescription of iron supplements during critical stages of pregnancy when it is needed most, and that pregnant women with food insecurity are more likely to have both lower iron intake and iron deficiency [53]. Last, prenatal iron supplementation could also be a contributing factor to iron status. Analyses of NHANES data from 1996 to 2006 [54] show that 77% of pregnant women self-reported taking a supplement containing iron (48 mg) in the previous 30 days, but previous studies have also found lower supplement use and iron intake in non-Hispanic Black pregnant women [55, 56]. Our work focused on iron status (concentrations and prevalences) rather than dietary sources such as iron supplementation, but based on prior evidence, we may infer that the majority of pregnant women in our analysis took an iron supplement during pregnancy.

The prevalence of ID in this study, based on total body iron, was 14%, which was similar to ID prevalence (also based on total body iron) observed in the Gupta et al., (16%) [7], and Mei et al., (18%) [9], both previous NHANES studies. The prevalence of anemia in our study was 8%, which was similar to the prevalence of 5% found by Mei et al. [9]. Our estimate of IDA was also similar to the 3% reported by Gupta et al. [7]. Slight discrepancies in prevalence rates between these studies may be explained by differing NHANES cycles included in analysis and differing age ranges. In the study by Gupta et al., the same cycles were used, but the age range (12–49 yrs.) was wider. The study by Mei et al. used fewer NHANES cycles (1999–2006) and included women aged 15–39 yrs. Overall, NHANES studies, including our current study, have observed that prevalence of ID is higher than prevalence of anemia and IDA. Total anemia prevalence (5–8%) in pregnant women is low in the US, compared with the global prevalence of 37% [57].

## Strengths & limitations

This study's main strength is that it uses a nationally representative sample of US pregnant women. Further, NHANES oversampled pregnant women during the 1999–2006 cycles and used comprehensive methods to assess iron status including the total body iron indicator. In addition, our sample size of 1,156 was large in comparison to the other US studies reporting associations between iron status and BMI during pregnancy [17, 18]. Our study has, however, some limitations. First, the cross-sectional design precludes establishing a temporal relationship. Second, as in any observational study, residual confounding may be present despite our effort to adjust our models, particularly due to not being able to include parity. Third, pre-

pregnancy weight was self-reported, which may misclassify maternal BMI. Fourth, due to the cross-sectional nature of the survey, we were unable to calculate gestational weight gain, which would have been more accurate in testing its association with iron status. Another potential confounder is gestational diabetes, which is only collected in two out of the six cycles included in this analysis, thus, we were unable to account for this factor. Fifth, the most recent iron status biomarker data combined with full pregnancy data (trimester or gestational age) in NHANES is over ten years old, and there is some evidence that it may overestimate the prevalence of anemia in low-income women [58]. Therefore, it is unknown whether these data are representative of the current population. NHANES should consider future collection of both iron status biomarkers and all variables needed to assess iron status during pregnancy. Last, the number of cases of anemia and especially of IDA was small in the overall sample and even smaller when stratified by BMI groups, making adjusted prevalence estimates less reliable and requiring caution for interpretation. Based on our *post hoc* power analysis, our sample size should have been adequate to detect clinically meaningful differences in ID, such as a 50% increase in prevalence (e.g. from 14% to 21%). However, our sample size did not have power to detect the very small observed differences.

## Conclusions

Overweight and obesity are common, and the prevalence continues to rise in pregnant women in the US. However, using the NHANES 1999–2010 cycles, we did not find women with pre-pregnancy overweight or obesity to have different iron status biomarker concentrations or an increased likelihood for ID and anemia during pregnancy when compared to pre-pregnancy normal weight. Previous studies reporting an association between BMI and iron status were conducted in different populations and may have been confounded by other demographic factors. Given these results and the fact that prenatal iron deficiency remains a significant public health concern, there is a need to continue collection of high-quality national data on these biomarkers during pregnancy to assess progress over time in alleviating maternal health disparities.

## Supporting information

**S1 File. This file contains three supplementary tables documenting 1) descriptive analysis of biomarkers, 2) prevalence of iron deficiency and anemia by race/ethnicity, and 3) a sensitivity analysis of prevalence by race/ethnicity.** S1 Table: Weighted medians and prevalences of inflammation, iron, and anemia biomarkers in pregnant women of reproductive age (18–49 years of age) from the NHANES (1999–2010) study (n = 1156). S2 Table: Weighted prevalences and prevalence ratios and differences of iron deficiency and anemia status in pregnant women ages 18–49 by race/ethnicity from NHANES 1999–2010 (n = 1156). S3 Table: Weighted prevalences and prevalence ratios and differences of iron deficiency and anemia status in pregnant women ages 18–49, by race/ethnicity from NHANES 1999–2010 (n = 1156) using Mexican American and Hispanic Other categories.
(DOCX)

## Acknowledgments

The authors would like to acknowledge the contribution of Elizabeth Soucy, who assisted with the initial literature search for this project.

## Author Contributions

**Conceptualization:** Mihaela A. Ciulei, Kelly Gallagher, Ruth A. Pobee, Rachel E. Walker.

**Data curation:** Mihaela A. Ciulei.

**Formal analysis:** Mihaela A. Ciulei, Djibril M. Ba, Celeste Beck, Rachel E. Walker.

**Funding acquisition:** Rachel E. Walker.

**Methodology:** Mihaela A. Ciulei, Kelly Gallagher, Djibril M. Ba, Celeste Beck, Ruth A. Pobee, Alison D. Gernand, Rachel E. Walker.

**Project administration:** Rachel E. Walker.

**Resources:** Alison D. Gernand.

**Supervision:** Alison D. Gernand, Rachel E. Walker.

**Visualization:** Rachel E. Walker.

**Writing – original draft:** Mihaela A. Ciulei, Kelly Gallagher, Djibril M. Ba, Celeste Beck, Ruth A. Pobee, Rachel E. Walker.

**Writing – review & editing:** Mihaela A. Ciulei, Kelly Gallagher, Djibril M. Ba, Celeste Beck, Ruth A. Pobee, Alison D. Gernand, Rachel E. Walker.

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
