## [Decision Letter · Decision Letter 0]

9 Apr 2024

PONE-D-24-08813Iron status in a representative sample of US pregnant women is not associated with pre-pregnancy BMI: results from the NHANES (1999-2010) studyPLOS ONE

Dear Dr. Walker,

Thank you for submitting your manuscript to PLOS ONE. After careful consideration, we feel that it has merit but does not fully meet PLOS ONE’s publication criteria as it currently stands. Therefore, we invite you to submit a revised version of the manuscript that addresses the points raised during the review process.

In addition to reviewers´ comments, please describe the procedures for selection of covariables used in the multiple adjusted models. One important concern is the small number of observations for the outcomes (anemia and iron deficiency anemia): this point should be discussed in relation to the limitations of sample power and external validity. 

We look forward to receiving your revised manuscript.

Kind regards,

Marly A. Cardoso, Ph.D.

Academic Editor

PLOS ONE

Journal Requirements:

Reviewers' comments:

Reviewer's Responses to Questions

**Comments to the Author**

1. Is the manuscript technically sound, and do the data support the conclusions?

Reviewer #1: Yes

Reviewer #2: No

2. Has the statistical analysis been performed appropriately and rigorously? 

Reviewer #1: Yes

Reviewer #2: Yes

3. Have the authors made all data underlying the findings in their manuscript fully available?

Reviewer #1: Yes

Reviewer #2: Yes

4. Is the manuscript presented in an intelligible fashion and written in standard English?

Reviewer #1: Yes

Reviewer #2: Yes

5. Review Comments to the Author

Reviewer #1: Thank you for the opportunity to read this interesting manuscript. I am pleased to see this paper that explores the association between pre-pregnancy BMI, iron status and anemia in a nationally representative sample of US pregnant women.

In my opinion, the manuscript is a good piece of work, well structured, clear and easy to follow and understand. The introduction section is well presented, provides a contextualization of the topic addressed and presents the justification for the study. The methods section provides sufficient information, is well detailed and can be replicated. The statistical analyses are sound and the results are robust. The conclusions are well-stated and, in the discussion section, the authors explain the results and make relevant comparisons with previous studies.

I have only 2 minor suggestions:

1. I suggest to include anemia in the title and objective of the study, since you also assessed this outcome in addition to iron status.

2. In the abstract (line 36), check the PR value for ID (overweight). It is different from what is shown in Table 2.

Reviewer #2: Reviewer Report:

This is an interesting study on Iron status among US pregnant women.

I propose to address the following minor issues.

Abstract:

Abstract was written properly.

Introduction:

Introduction is written properly.

Methods:

Study population: How the sample size was calculated? Is this sample sufficient to produce enough power?

Results:

It is better to put one figure to show the principle finding to make it easily understandable by the general reader.

Discussion:

Inflammation is predicted to be responsible for anemia among obese/overweight individuals, and CRP is a marker of inflammation. The author reported prevalence of women with CRP>0.5 mg/L. It was better to show the prevalence of women with iron deficiency based on the category of CRP.

6. PLOS authors have the option to publish the peer review history of their article (what does this mean?). If published, this will include your full peer review and any attached files.

Reviewer #1: No

Reviewer #2: **Yes: **Md Kamruzzaman

---

## [Author Response · Author response to Decision Letter 0]

10 May 2024

In addition to reviewers´ comments, please describe the procedures for selection of covariables used in the multiple adjusted models. 

Author comments: We selected covariables a priori based on prior literature of factors known to be related to iron and anemia status and BMI during pregnancy (please see the Methods/Covariates section, lines 136-140). 

One important concern is the small number of observations for the outcomes (anemia and iron deficiency anemia): this point should be discussed in relation to the limitations of sample power and external validity.

Author comments: Thank you for pointing out the small number of observations for the anemia and iron deficiency anemia outcomes. We followed the CDC guidelines and flagged the estimates that had a small sample size (<30). We thought the readers should see these results but interpret them with caution given the small sample size. We pointed this limitation out in the table footnotes, Results section (lines #237-240) and Strengths & Limitations section (lines #357-360). 

Reviewer #2 also had questions regarding our sample size and power, please see our answer under Methods (first question). 

Comments to the Author

Reviewer #1: Thank you for the opportunity to read this interesting manuscript. I am pleased to see this paper that explores the association between pre-pregnancy BMI, iron status and anemia in a nationally representative sample of US pregnant women.

In my opinion, the manuscript is a good piece of work, well structured, clear and easy to follow and understand. The introduction section is well presented, provides a contextualization of the topic addressed and presents the justification for the study. The methods section provides sufficient information, is well detailed and can be replicated. The statistical analyses are sound and the results are robust. The conclusions are well-stated and, in the discussion section, the authors explain the results and make relevant comparisons with previous studies.

I have only 2 minor suggestions:

1. I suggest to include anemia in the title and objective of the study, since you also assessed this outcome in addition to iron status. 

Author comments: Thank you for this helpful suggestion, which will provide more clarity to our readers regarding what we examined. We incorporated the term anemia in our title (lines 1-2), objective (line 75), and throughout the manuscript. 

2. In the abstract (line 36), check the PR value for ID (overweight). It is different from what is shown in Table 2.

Author comments: Thanks for spotting this discrepancy; it is now fixed. Line 37: “There were no differences in the prevalence of ID or anemia in women with pre-pregnancy overweight and obesity (ID: overweight, adjusted prevalence ratio (PR)=1.27, 95%CI: 0.89-1.82; obesity, PR=0.75, 95%CI: 0.39-1.45.”

In addition, we found that the p-values reported in the Figure 1 legend did not match the figure because they were from a previous version of the statistical analysis. We have updated the legend to be the correct numbers (lines 228-229).

Reviewer #2: Reviewer Report:

This is an interesting study on Iron status among US pregnant women.

I propose to address the following minor issues.

Abstract:

Abstract was written properly.

Introduction:

Introduction is written properly.

Methods:

Study population: How the sample size was calculated? Is this sample sufficient to produce enough power?

Author comments: Thanks for this question. As described in lines 81-91, our sample size was based on years of available NHANES data in which iron status biomarkers were measured. Importantly, our sample size of 1,156 was larger than other US studies that have investigated the association between BMI and iron status (Cao et al., 2016 and Dao et al., 2013). It is quite difficult to find studies of pregnant women with a larger sample size, especially a nationally representative sample. Using the Poisson regression option in G*power to conduct a post hoc power analysis, a sample size of 1,156 would give us >80% power to detect significant iron deficiency prevalence ratios of 1.5-2.0 (50-100% difference). However, we did not have enough power to detect the small observed differences between groups, especially with the low number of observed cases of anemia and IDA. We have added a description of the power calculation in the Methods (lines 91-96), Results (lines 240-251), and Strengths and Limitations (lines 360-363).

In our reporting, we followed the CDC guidelines (reference 47), which indicate that the effective number of cases should not be smaller than 30. We flagged such findings in our results. For instance, in Table 2, we indicate in the footnote that estimates from anemia and iron deficiency anemia in association with BMI categories may be unreliable due to a total number of cases less than 30. 

Results:

It is better to put one figure to show the principle finding to make it easily understandable by the general reader.

Author comments: Thank you for this comment – we agree that figures are important and help the reader easily understand the findings. In this case, we present the continuous biomarker results, the first part of our main findings, in a figure. However, we felt that the details of our main prevalence model results were best displayed in a table. While we could add another figure to the paper, we were following the general guidelines of not replicating the same results across tables and figures. 

Discussion:

Inflammation is predicted to be responsible for anemia among obese/overweight individuals, and CRP is a marker of inflammation. The author reported prevalence of women with CRP>0.5 mg/L. It was better to show the prevalence of women with iron deficiency based on the category of CRP.

Author comments: We welcome this valuable suggestion. We tested to see if inflammation (CRP>5 mg/L) was associated with cases of iron deficiency or IDA. We observed no differences by inflammatory status. This has been added to Table 1. Additionally, a prior publication by Mei et al., (reference #9) tested if inflammation impacts the prevalence of iron deficiency and anemia in pregnant women (used the same cycles as us but included a slightly different age range). Specifically, Mei et al., tested the results in the total sample and a subsample (removed pregnant women with CRP>5 mg/L) and found that the prevalence of ID in the subsample was not substantially different from the one from the total sample (reference #9). As such, our results are consistent with this prior publication.

---

## [Editor Report · Decision Letter 1]

14 May 2024

PONE-D-24-08813R1Iron status and anemia in a representative sample of US pregnant women is not associated with pre-pregnancy BMI: results from the NHANES (1999-2010) studyPLOS ONE

Dear Dr. Walker,

Thank you for submitting your manuscript to PLOS ONE. After careful consideration, we feel that it has merit but does not fully meet PLOS ONE’s publication criteria as it currently stands. Therefore, we invite you to submit a revised version of the manuscript that addresses the points raised during the review process.

The authors have made a good revision of the manuscript. However, an improvement on the description of the selection of covariates for multiple adjustment models is still necessary.  

We look forward to receiving your revised manuscript.

Kind regards,

Marly A. Cardoso, Ph.D.

Academic Editor

PLOS ONE

Journal Requirements:

Additional Editor Comments:

The authors have made a good revision of the manuscript. However, I suggest reviewing and providing details on how the covariates were selected for multiple models to avoid residual confounding. The authors have replied that "We selected covariables a priori based on prior literature of factors known to be related to iron and anemia status and BMI during pregnancy (please see the Methods/Covariates section, lines 136-140)." - saying just "based on prior literature" is not enough! There are many ways to test multiple adjustment models. I recommend the use of a hierarchical conceptual framework combined with different p-values for each determinant level. Please see the following recent paper: NCD Risk Factor Collaboration (NCD-RisC). Worldwide trends in underweight and obesity from 1990 to 2022: a pooled analysis of 3663 population-representative studies with 222 million children, adolescents, and adults. Lancet. 2024 Mar 16;403(10431):1027-1050. doi: 10.1016/S0140-6736(23)02750-2.

---

## [Author Response · Author response to Decision Letter 1]

4 Jun 2024

Thank you to the editor for thoughtful consideration of our manuscript. Below are our responses to the comments from the editor in response to our last revision. Author responses are below editor comments. All line numbers refer to the clean manuscript version.

Additional Editor Comments:

The authors have made a good revision of the manuscript. However, I suggest reviewing and providing details on how the covariates were selected for multiple models to avoid residual confounding. The authors have replied that "We selected covariables a priori based on prior literature of factors known to be related to iron and anemia status and BMI during pregnancy (please see the Methods/Covariates section, lines 136-140)." - saying just "based on prior literature" is not enough! There are many ways to test multiple adjustment models. I recommend the use of a hierarchical conceptual framework combined with different p-values for each determinant level. Please see the following recent paper: NCD Risk Factor Collaboration (NCD-RisC). Worldwide trends in underweight and obesity from 1990 to 2022: a pooled analysis of 3663 population-representative studies with 222 million children, adolescents, and adults. Lancet. 2024 Mar 16;403(10431):1027-1050. doi: 10.1016/S0140-6736(23)02750-2.

Author comments: Thank you for your comments and explanation. We have carefully reviewed the request and would like to note that there are significant differences between our methods and the paper that the editor kindly shared (doi: 10.1016/S0140-6736(23)02750-2). In the shared paper, researchers conducted a multi-level, longitudinal analysis using Bayesian models. Our analysis used a cross-sectional design (not hierarchical or multi-level) of a large dataset using complex survey design to obtain a nationally representative sample in the US (DHHS publication; nos. (PHS) 2013–1360 & (PHS) 2012–1355; https://wwwn.cdc.gov/nchs/nhanes/analyticguidelines.aspx#sample-design). The weighting procedures required for complex survey designs make it difficult to assess true correlation and collinearity between covariates within a model. We reviewed current practices for covariate identification and adjustment and see others in the field apply a similar approach to ours. Recent articles in PLOS One have used similar approaches to ours when conducting a similar analysis (e.g. doi: 10.1371/journal.pone.0303169). Additionally, other analyses of iron status with the NHANES data used the same or similar covariates (see: doi: 10.3945/ajcn.110.007195, 10.1371/journal.pone.0112216, and 0.3945/ajcn.117.155978). Therefore, we feel that our analysis is most comparable to these previously published reports when we include a similar list of covariates. For these reasons, we consider that our method of covariate selection was the appropriate approach in this case. We acknowledge that using hierarchical Bayesian criteria may be preferred when a dataset has a hierarchical or multi-level structure. We have updated our Methods section to better describe our approach and reference comparable literature in lines 136-139.

---

## [Editor Report · Decision Letter 2]

5 Jun 2024

PONE-D-24-08813R2Iron status and anemia in a representative sample of US pregnant women is not associated with pre-pregnancy BMI: results from the NHANES (1999-2010) studyPLOS ONE

Dear Dr. Walker,

Thank you for submitting your manuscript to PLOS ONE. After careful consideration, we feel that it has merit but does not fully meet PLOS ONE’s publication criteria as it currently stands. Therefore, we invite you to submit a revised version of the manuscript that addresses the points raised during the review process.

The revision of the procedures for data analysis needs improvement as recommended.  

We look forward to receiving your revised manuscript.

Kind regards,

Marly A. Cardoso, Ph.D.

Academic Editor

PLOS ONE

**Additional Editor Comments:**

The authors did not review the procedures for the selection of covariables in multiple regression models as recommended. In my previous comments, I have suggested a recent paper as an example of the use of hierarchical conceptual framework for selection of independent variables, independent of using Bayesian approach or any other statistical regression models. In the revised manuscript, the description of "we would expect to be associated with iron status based on prior literature" is not enough. There are so many methodological articles on this point and the authors must consider this revision. There is a classical paper on this (Victora CG, Huttly SR, Fuchs SC, Olinto MT. The role of conceptual frameworks in epidemiological analysis: a hierarchical approach. Int J Epidemiol. 1997 Feb;26(1):224-7. doi: 10.1093/ije/26.1.224) among many others combining conceptual frameworks with p-value and changes in the oucome measure estimates for selection of independent variables.

---

## [Author Response · Author response to Decision Letter 2]

16 Jul 2024

Additional Editor Comments:

The authors did not review the procedures for the selection of covariables in multiple regression models as recommended. In my previous comments, I have suggested a recent paper as an example of the use of hierarchical conceptual framework for selection of independent variables, independent of using Bayesian approach or any other statistical regression models. In the revised manuscript, the description of "we would expect to be associated with iron status based on prior literature" is not enough. There are so many methodological articles on this point and the authors must consider this revision. There is a classical paper on this (Victora CG, Huttly SR, Fuchs SC, Olinto MT. The role of conceptual frameworks in epidemiological analysis: a hierarchical approach. Int J Epidemiol. 1997 Feb;26(1):224-7. doi: 10.1093/ije/26.1.224) among many others combining conceptual frameworks with p-value and changes in the oucome measure estimates for selection of independent variables.

Author comments: Thank you for the opportunity to improve and resubmit this study. We have taken your recommendation to utilize a hierarchical conceptual framework for covariate selection. We have considered the methodology described by Victora et al., and have incorporated it into our statistical methods (see attached conceptual framework diagram). We describe this process in the Methods section (lines 182-199). Based on this process, we decided to remove family income from the adjusted models in this work, since it would likely be on the causal pathway between education and iron status outcomes and was not associated with iron outcomes in unadjusted models. Results, including tables and figures, have been revised accordingly. We hope you agree that this improves the interpretation of the results. Please let us know if you have any remaining questions about these adjusted models.

---

## [Editor Report · Decision Letter 3]

24 Jul 2024

Iron status and anemia in a representative sample of US pregnant women is not associated with pre-pregnancy BMI: results from the NHANES (1999-2010) study

PONE-D-24-08813R3

Dear Dr. Walker,

We’re pleased to inform you that your manuscript has been judged scientifically suitable for publication and will be formally accepted for publication once it meets all outstanding technical requirements.

Kind regards,

Marly A. Cardoso, Ph.D.

Academic Editor

PLOS ONE
---

## [Editor Report · Acceptance letter]

30 Jul 2024

PONE-D-24-08813R3 

PLOS ONE

Dear Dr. Walker, 

I'm pleased to inform you that your manuscript has been deemed suitable for publication in PLOS ONE. Congratulations! Your manuscript is now being handed over to our production team.

Kind regards, 

on behalf of

Dr. Marly A. Cardoso 

Academic Editor

PLOS ONE